# Complex Dynamics of a Cournot Quantum Duopoly Game with Memory and Heterogeneous Players

**DOI:** 10.3390/e24101333

**Published:** 2022-09-22

**Authors:** Luis Garcia-Perez, Juan Grau-Climent, Ramon Alonso-Sanz, Juan C. Losada

**Affiliations:** Complex Systems Group, Universidad Politécnica de Madrid, Ciudad Universitaria, 28040 Madrid, Spain

**Keywords:** quantum Cournot duopoly game, quantum entanglement, memory, heterogeneous players, local stability analysis, profit, bifurcation, chaos

## Abstract

Previous authors tend to consider a certain range of values of the parameters involved in a game, not taking into account other possible values. In this article, a quantum dynamical Cournot duopoly game with memory and heterogeneous players (one of them is boundedly rational and the other one, a naive player) is studied, where the quantum entanglement can be greater than one and the speed of adjustment can be negative. In this context, we analyzed the behavior of the local stability and the profit in those values. Considering the local stability, it is observed that the stability region increases in the model with memory regardless of whether the quantum entanglement is greater than one or whether the speed of adjustment is negative. However, it is also shown that the stability is greater in the negative than in the positive zone of the speed of adjustment and, therefore, it improves the results obtained in previous experiments. This increase of stability enables higher values of speed of adjustment and, as a result of that, the system reaches the stability faster, resulting in a remarkable economic advantage. Regarding the behavior of the profit with these parameters, the principal effect shown is that the application of memory causes a certain delay in the dynamics. Through this article, all these statements are analytically proved and widely supported with several numerical simulations, using different values of the memory factor, the quantum entanglement, and the speed of adjustment of the boundedly rational player.

## 1. Introduction

In the context of game theory, there is a common tendency to study a certain range of values of the parameters in the analysis of the models, without considering other possible ones. This paper intends to take into account some different values of the parameters involved in a game to find new scenarios which can lead to interesting results. This point of view can be relevant if we consider the variety of applications of game theory in fields such as economics, psychology, biology, etc. It may occur that a special value of a parameter in a game only has a sense in a specific science or subject of study, so that we would analyze the model partially if we did not take it into account. Taking this view as a starting point, we considered a wider range of values of some parameters involved in the Cournot duopoly game to find new cases which have not been studied before. As this analysis is focused on the economics, values with no economic meaning are excluded from the study. This paper can be seen as a next step in the researched previously started in [1]. Below, we briefly describe some common premises for both works, outlining the variations considered in this study. In economics, one of the typical market structures is the oligopoly, in which a few firms produce similar products. Cournot, in [2], presented a theoretical model of olygopoly, where the firms involved attempted to maximize profits by simultaneously choosing the amount of output to produce. Other models were proposed by authors such as Bertrand in [3], who proposed a duopoly model based on setting the prices of the players, or Stackelberg in [4] with a sequential model of the Cournot duopoly model. Another condition applicable to the game is the election of the player expectations, which defines the production of each firm in future periods. Each type of player adjusts the production for future periods to maximize the profit by using a different strategy. This article considers two players, one is a boundedly rational player and the other one with naive expectations in a similar way to [5,6], in the study of a nonlinear discrete-time Cournot duopoly game. The model of a boundedly rational player depends on a parameter, called speed of adjustment, which is usually positive; however, negative values of this parameter are also considered in this article.

Since the application of quantum game theory usually improves the results comparing to the classical games, it is also included in the game modelization. This technique was used for the first time in [7] and, through the years, many authors have carried on with this issue, highlighting the contribution of [8] where the two players of the Cournot duopoly virtually cooperate due to the quantum entanglement between them, since the quantity produced by each firm depends not only on the strategy of that player, but also on the strategy of the other one. In this context, it is shown that when the entanglement increases, the profits increase. As a consequence of that, a lot of articles related to the quantization of games have appeared, applying entanglement to different duopoly games such as the Cournot model in [9], the Bertrand model in [10], or the Stackelberg model in [11]. Other authors have analyzed this subject from a different point of view. In, e.g., [12], economic activity is described from a viewpoint of quantum games. The authors define a quantum commodity as a good which can be interchangeable between players to trade and it is studied how some of these goods, called quantum commodity money, can emerge as a media of exchange not to be consumed or used in production, under certain conditions. In order to compare the results with those described in [1,13], the model proposed in this paper follows the Li–Du–Massar scheme proposed in [8]. The difference is that not only values between zero and one of the quantum entanglement are considered, but also values greater than one of this parameter are analyzed. The maximum quantum entanglement is limited by the conditions required to verify the economic meaning of the game, as is explained in the following sections.

Our model implements a model of memory based on an average of all the past states with geometric decay, since the results obtained previously are better compared to the system in absence of memory. Most of the works with memory are related with the popular logistic map [14]. The pioneers in these studies with the logistic map were [15,16]. Before this paper, we investigated the effect of memory in systems that are discrete in all their components (space, time, and state variable), i.e., cellular automata, in [17], as well as in the logistic map in [18,19].

Finally, at the end of the paper, there is a study of the behavior of the profit for the two players with and without memory and applying different values of the quantum entanglement and the speed of adjustment of the boundedly rational player. This analysis is focused on searching an economic sense to the model under different conditions to understand the variations observed as a result of that.

This paper is organized as follows. In Section 2, a dynamical quantum Cournot duopoly game model with heterogeneous players is briefly described. The equilibrium points, as well as their local stability conditions, and the relationship between quantum entanglement and stability region are also studied in this section, with special emphasis on the new conditions, i.e., speed of adjustment less than zero and degree of entanglement greater than one. In Section 3, numerical simulation is used to show the dynamics of the system and to support the results obtained in the previous section. Section 4 analyzes the behavior of the profit with and without memory with different values of the quantum entanglement and the speed of adjustment of the boundedly rational player. Finally, Section 5 presents the conclusions and meaning of this paper.

## 2. The Model

Based on the article [13], we consider the classical Cournot’s duopoly in which two firms are producing perfect substitute goods in a duopolistic market. The cost function is the same for both firms and is taken in linear form: $C_{i}\left(q_{i}\left(t\right)\right)=c_{i}q_{i}\left(t\right)$. This classical Cournot’s duopoly competition is taken in the quantum domain with the use of the Li–Du–Massar entanglement structure based on quantum methods for continuous-variable quantum games. We apply this model of quantization but there are other examples of similar models, such as EWL or MW schemes [20]. Specifically, Eisert et al. described the EWL model in [21] and applied it to the prisoners’ dilemma, showing that this game ceases to pose a dilemma if quantum strategies are allowed for. We used the LDM model of quantization to compare the results with those described in [1,13], since it has been applied successfully by other authors. This entangled model has several steps. First, the game starts from *initial state*|00〉. This state undergoes a unitary entanglement operation $\hat{J}\left(\gamma \right)=e^{-\gamma \left({\hat{a}}_{1}^{†}{\hat{a}}_{2}^{†}-{\hat{a}}_{1}{\hat{a}}_{2}\right)}$, where $a_{i}^{†}\left({\hat{a}}_{i}\right)$ represents the creation (annihilation) operator of the firm’s *i* electromagnetic field and γ≥0 is known as the *squeezing parameter* and can be reasonably regarded as a measure of entanglement. Next, the two firms execute their *strategic moves* via unitary operation ${\hat{D}}_{i}\left(x_{i}\right)=e^{x_{i}\left({\hat{a}}_{i}^{†}-{\hat{a}}_{i}\right)/\sqrt{2}},i=1,2$. Finally, these two firms’ states are measured after *a disentanglement operation*
$\hat{J}{\left(\gamma \right)}^{†}$. Thus, the *final state* is carried out by $|\psi_{f}\rangle =\hat{J}{\left(\gamma \right)}^{†}\left({\hat{D}}_{1}\left(x_{1}\right)⊗{\hat{D}}_{2}\left(x_{2}\right)\right)\hat{J}\left(\gamma \right)|00\rangle$. The final measurement gives the respective quantum quantities of the two firms:(1)$$\begin{array}{c} q_{1}^{c}=x_{1}\cosh\gamma +x_{2}\sinh\gamma ,\\ q_{2}^{c}=x_{2}\cosh\gamma +x_{1}\sinh\gamma , \end{array}$$
where $x_{1}$ and $x_{2}$ represent the independent quantities and $q_{1}^{c}$ and $q_{2}^{c}$ are the entangled quantities used by the two firms in the quantum game. When the degree of entanglement is zero, i.e., γ=0, then the quantum game turns into the original classic form and $q_{i}^{c}\left(t\right)=x_{i}\left(t\right)=q_{i}\left(t\right)$. Therefore, the market quantum price, based on the linear inverse demand function, is
(2)$$p^{c}=a-bQ^{c}=a-be^{\gamma }\left(x_{1}+x_{2}\right),$$
being $Q^{c}=q_{1}^{c}+q_{2}^{c}$ and a>0 and b>0 and $a>bQ^{c}$. Then, we can find that the dynamical quantum profits of the two firms are
(3)$$\left\{\begin{array}{c} u_{1}^{c}\left(t\right)=q_{1}^{c}\left(t\right)\left(p^{c}\left(t\right)-c_{1}\right)=\left(x_{1}\left(t\right)\cosh\gamma +x_{2}\left(t\right)\sinh\gamma \right)\left(a-be^{\gamma }\left(x_{1}\left(t\right)+x_{2}\left(t\right)\right)-c_{1}\right),\\ u_{2}^{c}\left(t\right)=q_{2}^{c}\left(t\right)\left(p^{c}\left(t\right)-c_{2}\right)=\left(x_{2}\left(t\right)\cosh\gamma +x_{1}\left(t\right)\sinh\gamma \right)\left(a-be^{\gamma }\left(x_{1}\left(t\right)+x_{2}\left(t\right)\right)-c_{2}\right). \end{array}\right.$$
We also consider this study with different expectations (heterogeneous expectations), the first player being a boundedly rational player, and the other one a naive player. Since the knowledge of the market is usually incomplete, the first firm with boundedly rational expectations builds his output decision on the basis of the estimation of the marginal profit $\frac{\partial u_{1}^{c}\left(t\right)}{\partial x_{1}\left(t\right)}$. This player decides to increase or decrease the production if the marginal profit is positive or negative, respectively. Then, this boundedly rational player can be modeled as
(4)$$x_{1}\left(t+1\right)=x_{1}\left(t\right)+\alpha x_{1}\left(t\right)\frac{\partial u_{1}^{c}\left(t\right)}{\partial x_{1}\left(t\right)}=x_{1}\left(t\right)+\alpha x_{1}\left(t\right)\left(\left(a-c_{1}\right)\cosh\gamma -b\left(1+e^{2\gamma }\right)x_{1}\left(t\right)-be^{2\gamma }x_{2}\left(t\right)\right),$$
where α is a parameter which represents the speed of adjustment.

The independent quantity of the naive player maximizes the expected profit. To obtain the maximum benefit we have to derive $u_{2}^{c}\left(t\right)$ with respect to $x_{2}\left(t\right)$ and equalize to 0 to resolve $x_{2}^{*}\left(t\right)$; then, the naive player can be modeled as follows:(5)$$x_{2}\left(t+1\right)=x_{2}^{*}\left(t\right)=\frac{\left(a-c_{2}\right)\cosh\gamma -be^{2\gamma }x_{1}\left(t\right)}{b\left(1+e^{2\gamma }\right)}.$$
It is shown that it is really a maximum because it verifies the second-order condition, i.e.,
(6)$$\frac{\partial^{2}u_{2}^{c}\left(t\right)}{\partial x_{2}{\left(t\right)}^{2}}=-b\left(1+e^{2\gamma }\right)<0$$
recalling that *b* is a positive constant.

Based on the article [13], we also add the memory effect studied earlier in these articles [17,18,19]. We consider a weighted average memory with geometric decay $\mu_{i}\left(t\right)$, which is defined in the article [19] as
(7)$$\mu_{i}\left(t\right)=\frac{x_{i}\left(t\right)+\sum_{j=1}^{t-1}\beta^{t-j}x_{i}\left(j\right)}{1+\sum_{j=1}^{t-1}\beta^{t-j}}\equiv \frac{\omega_{i}\left(t\right)}{\Omega_{i}\left(t\right)}\;\mathrm{for}\;t>1,$$
being,
(8)$$\omega_{i}\left(t+1\right)=x_{i}\left(t+1\right)+\beta \omega_{i}\left(t\right),$$
and,
(9)$$\Omega_{i}\left(t+1\right)=1+\beta \Omega_{i}\left(t\right).$$
where β represents the memory factor and $\mu_{i}\left(1\right)=x_{i}\left(1\right)$ and Ω(1)=1. The limit case β=1 corresponds to a memory with equally weighted records (*full* memory), whereas β≪1 intensifies the contribution of the most recent states (short-term working memory). The choice β=0 leads to the memoryless model. Therefore, the dynamical quantum Cournot duopoly game with heterogeneous players with memory can be described using the following two-dimensional discrete-time dynamical system:(10)$$\left\{\begin{array}{cc} x_{1}\left(t+1\right) & =\mu_{1}\left(t\right)+\alpha^{\prime }\mu_{1}\left(t\right)\left(\left(a-c_{1}\right)\cosh\gamma -b\left(1+e^{2\gamma }\right)\mu_{1}\left(t\right)-be^{2\gamma }\mu_{2}\left(t\right)\right),\\ x_{2}\left(t+1\right) & =\frac{\left(a-c_{2}\right)\cosh\gamma -be^{2\gamma }\mu_{1}\left(t\right)}{b\left(1+e^{2\gamma }\right)}, \end{array}\right.$$
where $\alpha^{\prime }$ is a parameter which represents the speed of adjustment in the game with memory for the first player.

As $x_{i},i=1,2$ are the independent quantities, they must have a positive value and, if at any step, the result of the equation is negative, we consider that this quantity $x_{i}$ is zero.

### 2.1. Equilibrium Points

Considering, for long times and β<1, $\mu \left(t\right)=\omega \left(t\right)\left(1-\beta \right)$, the two-dimensional discrete-time dynamical system of Equation (Equation 10) can be written as
(11)$$\left\{\begin{array}{cc} \omega_{1}\left(t+1\right) & =\omega_{1}\left(t\right)+\alpha^{\prime }\omega_{1}\left(t\right)\left(1-\beta \right)\left(\left(a-c_{1}\right)\cosh\gamma -b\left(1+e^{2\gamma }\right)\omega_{1}\left(t\right)\left(1-\beta \right)-be^{2\gamma }\omega_{2}\left(t\right)\left(1-\beta \right)\right),\\ \omega_{2}\left(t+1\right) & =\beta \omega_{2}\left(t\right)+\frac{\left(a-c_{2}\right)\cosh\gamma -be^{2\gamma }\omega_{1}\left(t\right)\left(1-\beta \right)}{b\left(1+e^{2\gamma }\right)}. \end{array}\right.$$
To find the equilibrium points of Equation (Equation 11), we can replace all $\omega_{i}\left(t+1\right)$ and $\omega_{i}\left(t\right)$ for $\omega_{i}$ into Equation (Equation 11), obtaining the following nonlinear algebraic system:(12)$$\left\{\begin{array}{c} \omega_{1}\left(1-\beta \right)\left(\left(a-c_{1}\right)\cosh\gamma -b\left(1+e^{2\gamma }\right)\omega_{1}\left(1-\beta \right)-be^{2\gamma }\omega_{2}\left(1-\beta \right)\right)=0,\\ \frac{\left(a-c_{2}\right)\cosh\gamma -be^{2\gamma }\omega_{1}\left(1-\beta \right)}{b\left(1+e^{2\gamma }\right)\left(1-\beta \right)}-\omega_{2}=0. \end{array}\right.$$

In Equation (Equation 12) there are two fixed points. As can be seen in the first equation, one solution for player 1 is given by $\omega_{1}^{I}=x_{1}^{I}=0$ (where, for a long time, $\mu_{i}=x_{i}$), and then substituting into the second equation, we have for player 2
(13)$$\omega_{2}^{I}=\frac{\left(a-c_{2}\right)\cosh\gamma }{b\left(1+e^{2\gamma }\right)\left(1-\beta \right)},$$
due to $\omega_{2}^{I}=\frac{\mu_{2}^{I}}{\left(1-\beta \right)}$ and based on the article [1], where for long times
(14)$$\mu_{i}\left(t\right)=\frac{x_{i}\left(t\right)\left(1+\sum_{j=1}^{t-1}\beta^{t-j}\right)}{1+\sum_{j=1}^{t-1}\beta^{t-j}}=x_{i}\left(t\right),$$
it is obtained:(15)$$\mu_{2}^{I}=x_{2}^{I}=\frac{\left(a-c_{2}\right)\cosh\gamma }{b\left(1+e^{2\gamma }\right)}.$$

Then, the first fixed point of the system is $E^{I}=\left(0,\frac{\left(a-c_{2}\right)\cosh\gamma }{b\left(1+e^{2\gamma }\right)}\right)$.

The other fixed point is determined by these equations:(16)$$\left\{\begin{array}{c} \left(a-c_{1}\right)\cosh\gamma -b\left(1+e^{2\gamma }\right)\omega_{1}^{II}\left(1-\beta \right)-be^{2\gamma }\omega_{2}^{II}\left(1-\beta \right)=0,\\ \frac{\left(a-c_{2}\right)\cosh\gamma -be^{2\gamma }\omega_{1}^{II}\left(1-\beta \right)}{b\left(1+e^{2\gamma }\right)\left(1-\beta \right)}-\omega_{2}^{II}=0. \end{array}\right.$$

Thus, the second fixed point of the system is $E^{II}=\left(x_{1}^{II},x_{2}^{II}\right)$, where
(17)$$\omega_{1}^{II}=\frac{\left(a-c_{1}+e^{2\gamma }\left(c_{2}-c_{1}\right)\right)\cosh\gamma }{b\left(1+2e^{2\gamma }\right)\left(1-\beta \right)},\omega_{2}^{II}=\frac{\left(a-c_{2}+e^{2\gamma }\left(c_{1}-c_{2}\right)\right)\cosh\gamma }{b\left(1+2e^{2\gamma }\right)\left(1-\beta \right)},$$
and so $\omega^{II}=\frac{\mu^{II}}{\left(1-\beta \right)}$, then:(18)$$\mu_{1}^{II}=x_{1}^{II}=\frac{\left(a-c_{1}+e^{2\gamma }\left(c_{2}-c_{1}\right)\right)\cosh\gamma }{b\left(1+2e^{2\gamma }\right)},\mu_{2}^{II}=x_{2}^{II}=\frac{\left(a-c_{2}+e^{2\gamma }\left(c_{1}-c_{2}\right)\right)\cosh\gamma }{b\left(1+2e^{2\gamma }\right)}.$$

Since the quantum Nash equilibrium has economic meaning and the fixed point $E^{II}$ is stable when α>0, as is shown in [1], then it must verify that
(19)$$\left\{\begin{array}{cc} a-c_{1}+e^{2\gamma }\left(c_{2}-c_{1}\right) & >0,\\ a-c_{2}+e^{2\gamma }\left(c_{1}-c_{2}\right) & >0. \end{array}\right.$$

If we set these values
(20)$$m=\frac{c_{1}-c_{2}}{a-c_{1}},k=a-c_{1},$$
then Equation (Equation 19) can be expressed as
(21)$$\left\{\begin{array}{cc} 1-me^{2\gamma } & >0,\\ 1+m+me^{2\gamma } & >0, \end{array}\right.$$
where *m* represents the relative marginal cost difference.

### 2.2. Local Stability

In this section, we analyze the local stability of the equilibrium points for all the values of α, positive and negative, taking into account the new conditions, $\alpha^{\prime }<0$ and γ>1, and considering two different zones in this study: m≤0 and m>0. When $\alpha^{\prime }\geq 0$, as is shown in [1], the fixed point $E^{II}$ is stable and gives the stability condition of the system:(22)$$\alpha^{\prime }<\frac{2\left(1+\beta \right)\left(1+e^{2\gamma }\right)\left(1+2e^{2\gamma }\right)}{\left(1-\beta \right)\left[{\left(1+e^{2\gamma }\right)}^{2}\left(1+\beta \right)+e^{4\gamma }\left(1-\beta \right)\right]\left[a-c_{1}+e^{2\gamma }\left(c_{2}-c_{1}\right)\right]\cosh\gamma },$$
with the equivalences mentioned in Equation (Equation 20), we can simplify as follows:(23)$$\alpha^{\prime }<\frac{2\left(1+\beta \right)\left(1+e^{2\gamma }\right)\left(1+2e^{2\gamma }\right)}{k\left(1-\beta \right)\left[{\left(1+e^{2\gamma }\right)}^{2}\left(1+\beta \right)+e^{4\gamma }\left(1-\beta \right)\right]\left(1-me^{2\gamma }\right)\cosh\gamma }.$$

To study the stability when $\alpha^{\prime }<0$, it is necessary to usethe Jacobian matrix J of Equation (Equation 11), which can be expressed as
(24)$$J\left(\omega_{1},\omega_{2}\right)=\left(\begin{array}{cc} 1+\alpha^{\prime }\left(1-\beta \right)\left[\left(a-c_{1}\right)\cosh\gamma -\left(1-\beta \right)\left(2b\left(1+e^{2\gamma }\right)\omega_{1}-be^{2\gamma }\omega_{2}\right)\right] & -\alpha^{\prime }be^{2\gamma }{\left(1-\beta \right)}^{2}\omega_{1}\\ -\frac{e^{2\gamma }}{1+e^{2\gamma }}\left(1-\beta \right) & \beta \end{array}\right).$$
We start with the analysis of the stability of the second equilibrium point $E^{II}$. The Jacobian matrix Equation (Equation 24) at this point can be written as
(25)$$J\left(E^{II}\right)=\left(\begin{array}{cc} 1-\alpha^{\prime }b{\left(1-\beta \right)}^{2}\left(1+e^{2\gamma }\right)\omega_{1}^{II} & -\alpha^{\prime }be^{2\gamma }{\left(1-\beta \right)}^{2}\omega_{1}^{II}\\ -\frac{e^{2\gamma }}{1+e^{2\gamma }}\left(1-\beta \right) & \beta \end{array}\right).$$

The characteristic polynomial can be expressed as
(26)$$p\left(\lambda \right)=\lambda^{2}-Tr\left(J\right)\lambda +Det\left(J\right),$$
where $Tr\left(J\right)$ and $Det\left(J\right)$ represent the trace and the determinant of the Jacobian matrix $J\left(E^{II}\right)$ and are given by
(27)$$\begin{array}{cc} Tr\left(J\right) & =1+\beta -\alpha^{\prime }b{\left(1-\beta \right)}^{2}\left(1+e^{2\gamma }\right)\omega_{1}^{II},\\ Det\left(J\right) & =\beta -\alpha^{\prime }b\beta {\left(1-\beta \right)}^{2}\left(1+e^{2\gamma }\right)\omega_{1}^{II}-\frac{\alpha^{\prime }b{\left(1-\beta \right)}^{3}e^{4\gamma }\omega_{1}^{II}}{1+e^{2\gamma }}. \end{array}$$

At this point, the Jury criterion can be applied to analyze the local stability of the second equilibrium point $E^{II}$, checking these three conditions:(28)$$\left\{\begin{array}{c} \left(a\right)\;p\left(1\right)=1-Tr\left(J\right)+Det\left(J\right)>0,\\ \left(b\right)\;p\left(-1\right)=1+Tr\left(J\right)+Det\left(J\right)>0,\\ \left(c\right)\;Det\left(J\right)<1. \end{array}\right.$$

The first condition can be written as follows, taking into account Equation (Equation 27)
(29)$$1-Tr\left(J\right)+Det\left(J\right)=\frac{\alpha^{\prime }b{\left(1-\beta \right)}^{3}\omega_{1}^{II}\left[{\left(1+e^{2\gamma }\right)}^{2}-e^{4\gamma }\right]}{1+e^{2\gamma }}=\frac{\alpha^{\prime }b{\left(1-\beta \right)}^{3}\omega_{1}^{II}\left(1+2e^{2\gamma }\right)}{1+e^{2\gamma }}>0,$$
which is never satisfied when $\alpha^{\prime }<0$. Then, it is not necessary to verify the other conditions to prove that the fixed point $E^{II}$ is unstable for $\alpha^{\prime }<0$. Now, considering the first equilibrium point $E^{I}$, the Jacobian matrix Equation (Equation 24) in this point can be calculated as
(30)$$J\left(E^{I}\right)=\left(\begin{array}{cc} 1+\alpha^{\prime }\left(1-\beta \right)\frac{\left(a-c_{1}+e^{2\gamma }\left(c_{2}-c_{1}\right)\right)\cosh\gamma }{\left(1+e^{2\gamma }\right)} & 0\\ -\frac{e^{2\gamma }}{1+e^{2\gamma }}\left(1-\beta \right) & \beta \end{array}\right).$$

There are two eigenvalues of the Jacobian matrix $J(E^{I})$:(31)$$\lambda_{1}=1+\alpha^{\prime }\left(1-\beta \right)\frac{\left(a-c_{1}+e^{2\gamma }\left(c_{2}-c_{1}\right)\right)\cosh\gamma }{\left(1+e^{2\gamma }\right)}=1+\alpha^{\prime }\left(1-\beta \right)\frac{k\left(1-me^{2\gamma }\right)cosh\gamma }{\left(1+e^{2\gamma }\right)},$$
(32)$$\lambda_{2}=\beta$$
The equilibrium point $E^{I}$ will be stable if the modulus of both eigenvalues are less than 1. Since β<1 is one of the assumptions mentioned previously in the model, it is clear that $\left|\lambda_{2}\right|<1$. Considering the eigenvalue $\lambda_{1}$, the condition $\left|\lambda_{1}\right|<1$ is satisfied for $\alpha^{\prime }<0$ when it verifies
(33)$$-1<1+\alpha^{\prime }\left(1-\beta \right)\frac{\left(a-c_{1}+e^{2\gamma }\left(c_{2}-c_{1}\right)\right)\cosh\gamma }{\left(1+e^{2\gamma }\right)}<1,$$
(34)$$-2<\alpha^{\prime }\left(1-\beta \right)\frac{\left(a-c_{1}+e^{2\gamma }\left(c_{2}-c_{1}\right)\right)\cosh\gamma }{\left(1+e^{2\gamma }\right)}<0.$$

This compound inequality can be divided into two parts and it results in
(35)$$\alpha^{\prime }\left(1-\beta \right)\frac{\left(a-c_{1}+e^{2\gamma }\left(c_{2}-c_{1}\right)\right)\cosh\gamma }{\left(1+e^{2\gamma }\right)}<0,$$
(36)$$\alpha^{\prime }\left(1-\beta \right)\frac{\left(a-c_{1}+e^{2\gamma }\left(c_{2}-c_{1}\right)\right)\cosh\gamma }{\left(1+e^{2\gamma }\right)}>-2.$$

When $\alpha^{\prime }<0$, according to the condition β<1, Equation (Equation 35) is always verified if
(37)$$\left(a-c_{1}+e^{2\gamma }\left(c_{2}-c_{1}\right)\right)>0,$$
or for the same, $1-me^{2\gamma }>0$. Additionally, if we isolate $\alpha^{\prime }$ in Equation (Equation 36), then we have
(38)$$\alpha^{\prime }>\frac{-2\left(1+e^{2\gamma }\right)}{\left(1-\beta \right)\left(a-c_{1}+e^{2\gamma }\left(c_{2}-c_{1}\right)\right)\cosh\gamma }.\;$$

Using the equivalences mentioned in Equation (Equation 20), we can simplify Equation (Equation 38) as follows:(39)$$\alpha^{\prime }>\frac{-2\left(1+e^{2\gamma }\right)}{k\left(1-\beta \right)\left(1-me^{2\gamma }\right)\cosh\gamma }.$$

Therefore, the first equilibrium point $E^{I}$ is stable when Equation (Equation 38) is verified, in contrast to the case $\alpha^{\prime }>0$, described previously in [1], where this fixed point is unstable. It is also proved that, having $\alpha^{\prime }<0$, the stability region is bigger when β increases.

Since in the non-memory model with α<0 the stability condition is $\alpha >\frac{-2\left(1+e^{2\gamma }\right)}{k\left(1-me^{2\gamma }\right)\cosh\gamma }$, we can express $\alpha^{\prime }$ as a function of α, as follows:(40)$$\alpha^{\prime }\geq \frac{\alpha }{1-\beta }.$$
Thus, having α<0 and $\alpha^{\prime }<0$, it is proved that the stability region in the game with memory is bigger than in the non-memory game in every value of the quantum entanglement γ, assuming the same conditions. As a conclusion, it can be stated that the system is stable for the values of $\alpha^{\prime }$ which verify the following expression:(41)$$\frac{-2\left(1+e^{2\gamma }\right)}{k\left(1-\beta \right)\left(1-me^{2\gamma }\right)\cosh\gamma }<\alpha^{\prime }<\frac{2\left(1+\beta \right)\left(1+e^{2\gamma }\right)\left(1+2e^{2\gamma }\right)}{k\left(1-\beta \right)\left[{\left(1+e^{2\gamma }\right)}^{2}\left(1+\beta \right)+e^{4\gamma }\left(1-\beta \right)\right]\left(1-me^{2\gamma }\right)cosh\gamma }$$
This expression can be simplified as follows:(42)$$-\alpha_{i}<\alpha^{\prime }<\delta \left(\gamma ,\beta \right)\alpha_{i},$$
where $\alpha_{i}$ and δ(γ,β) can be expressed as
(43)$$\alpha_{i}=\frac{2\left(1+e^{2\gamma }\right)}{k\left(1-\beta \right)\left(1-me^{2\gamma }\right)\cosh\gamma },$$
(44)$$\delta \left(\gamma ,\beta \right)=\frac{\left(1+\beta \right)\left(1+2e^{2\gamma }\right)}{{\left(1+e^{2\gamma }\right)}^{2}\left(1+\beta \right)+e^{4\gamma }\left(1-\beta \right)}.\;$$
It is shown that the stability is greater in the negative than in the positive zone because it always verifies that δ(γ,β)<1, independently of the value of γ and β considered.

It is important to obtain the values of *m* and γ where this study is valid (economic meaning). In $\alpha^{\prime }>0$, as we mentioned in Equation (Equation 21), due to the economic meaning of the Nash equilibrium, the expressions $1-me^{2\gamma }$ and $1+m+me^{2\gamma }$ must be positive. On one side, $1-me^{2\gamma }>0$ is always verified when m≤0 and γ≥0 but, in contrast, when m>0, this expression can be positive, negative, or zero depending on the value of γ. At this point, we calculate the value of γ where $1-me^{2\gamma }=0$ to establish the limit between the negative and positive values of this parameter. This particular value of γ, called $\gamma_{0}$, can be expressed as follows:(45)$$\gamma_{0}=\frac{1}{2}ln\left(\frac{1}{m}\right)\;\mathrm{when}\;m>0.$$

From $\gamma_{0}$, we can deduce the positive and negative region of $1-me^{2\gamma }$ when m>0:(46)$$\left\{\begin{array}{cc} 1-me^{2\gamma }>0 & 0\leq \gamma <\gamma_{0},\\ 1-me^{2\gamma }\leq 0 & \gamma \geq \gamma_{0}. \end{array}\right.$$

On the other side, $1+m+me^{2\gamma }>0$ is always verified when m>0, but, on the contrary, it does not happen when m≤0. Then, we obtain the value of γ, called $\gamma_{1}$, where $1+m+me^{2\gamma }=0$, as we obtain in the previous case:(47)$$\gamma_{1}=\frac{1}{2}ln\left(\frac{-1-m}{m}\right)\;\mathrm{when}\;m\leq 0.$$

From $\gamma_{1}$, we can deduce the positive and negative region of $1+m+me^{2\gamma }$ when m≤0:(48)$$\left\{\begin{array}{cc} 1+m+me^{2\gamma }>0 & 0\leq \gamma <\gamma_{1},\\ 1+m+me^{2\gamma }\leq 0 & \gamma \geq \gamma_{1}. \end{array}\right.$$

Therefore, in $\alpha^{\prime }>0$, the study of the stability of this section is only valid for these values:(49)$$\left\{\begin{array}{cc} m>0 & 0\leq \gamma <\gamma_{0},\\ m\leq 0 & 0\leq \gamma <\gamma_{1}. \end{array}\right.$$

In $\alpha^{\prime }<0$, to ensure that the Nash equilibrium point $E_{1}$ has economic meaning, the value of Equation (Equation 15) must be positive. As b>0 and $1+e^{2\gamma }>0$, it verifies that the denominator is always positive. Considering the numerator of the equation, since coshγ>0 when γ≥0, then it must be $a-c_{2}>0$ or forthe same $a>c_{2}$, which is always true. As is shown in Equation (Equation 37), the condition $1-me^{2\gamma }>0$ is necessary for the stability of $E^{I}$. This condition is always verified when m≤0 and γ≥0, but, in contrast, when m>0 this expression can be positive, negative, or zero depending on the value of γ. At this point, as we calculated previously, the value of γ where $1-me^{2\gamma }=0$ establishes the limit between the negative and positive values of this expression. This particular value of γ, called $\gamma_{0}$, is expressed in Equation (Equation 45). Therefore, in $\alpha^{\prime }<0$, the study of the stability of this section is only valid for these values:(50)$$\left\{\begin{array}{cc} m>0 & 0\leq \gamma <\gamma_{0},\\ m\leq 0 & \mathrm{in}\;\mathrm{all}\;\gamma . \end{array}\right.$$

## 3. Numerical Simulation

On this basis, we use numerical simulation to study the effect of memory in the system and graphically support the results analytically obtained throughout this paper. We consider the previous simulations carried out in [1] as a starting point and, as is described within, two different zones are included in this study: m>0 and m≤0. Under these premises, memory is applied to the quantum game to observe the variation of the stability and dynamical behavior of the system, with special attention given to the values $\alpha^{\prime }<0$ and γ>1.

At first, the region m>0 is analyzed, considering the value m=0.1 (we set $a=10,\;b=0.5,\;c_{1}=3,and\;c_{2}=2.3$) without memory in Figure 1 and with several values of the memory factor, β (0, 0.2, 0.5, and 0.8) in Figure 2. These figures show that the stability is greater in negative values of $\alpha^{\prime }$ than in positive ones for every specific value of γ and β, and the stability region increases with β, both in negative and positive values of $\alpha^{\prime }$.

Due to its mathematical importance, the region without economic meaning is also represented by a blue shaded area, i.e., $\gamma \geq \gamma_{0}$, where $\gamma_{0}=1.15$ in this example, according to Equation (Equation 45). Apart from being the edge between both regions, there is an asymptote in $\gamma_{0}$, which can be deduced from the inequality Equation (Equation 41) because the denominators cancel out since it verifies $1-me^{2}\gamma =0$ in this point. It is relevant that the equilibrium of the fixed points is inverted when $\gamma >\gamma_{0}$ due to the sign change of the expression $1-me^{2}\gamma$, as is shown in Equation (Equation 46).

In these figures, it is also shown that for negative values of $\alpha^{\prime }$, if γ increases, the stability zone is greater because the modulus of $\alpha^{\prime }$ increases up to this asymptote in $\gamma_{0}$, and it has the same behavior than the case of positive values of $\alpha^{\prime }$. This verifies up to a point because there is an exception when the value of the relative marginal cost difference *m* is between 0 and 0.1. As can be seen in Figure 3 with the example m=0.014, there is a range of values of γ where $\alpha^{\prime }$ decreases when γ increases, which means that the stability zone is smaller in these values. Likewise, Figure 4 shows that from m=0.0 to m=0.1 there is a range of values of γ where $\alpha^{\prime }$ decreases when γ increases.

The bifurcation diagrams of output of the first player are also represented for the values γ=0 and γ=0.7 in the region m>0, considering m=0.1 (we set a=10, b=0.5, $c_{1}=3,\;and\;c_{2}=2.3$) in Figure 5 and Figure 6. In these figures we show the increase of stability for $\alpha^{\prime }<0$ when the value of β is greater, as well as for $\alpha^{\prime }>0$.

Secondly, the local stability in the region m≤0 for m=−0.1 (we set $a=10,\;b=0.5,\;c_{1}=3,\;c_{2}=3.7$) without memory is represented in Figure 7 and with several values of the memory factor, β (0, 0.2, 0.5, and 0.8) in Figure 8. As we previously mentioned, it can be seen that the stability is always greater in the negative zone for every γ and β considered, and the stability region increases with β, both in negative and positive values of $\alpha^{\prime }$.

Similarly to the previous region of study, the zone without economic meaning, $\gamma \geq \gamma_{1}$ where $\gamma_{1}=1.09$ considering Equation (Equation 47), is also represented in the figures because it is interesting from a mathematical point of view. In fact, a minimum in the negative zone of $\alpha^{\prime }$ can be observed, which corresponds to a stability maximum in this zone. To calculate the value of γ for this maximum, we can substitute the expression $\cosh\gamma =\frac{e^{\gamma }+e^{-\gamma }}{2}$ into Equation (Equation 39), and simplifying, we obtain
(51)$$\alpha^{\prime }>f\left(\gamma \right)=\frac{-4e^{\gamma }}{k\left(1-\beta \right)\left(1-me^{2\gamma }\right)}.$$

Then, the stability maximum is given by the derivative of the right term of the inequality equals zero:(52)$$\frac{\partial f(\gamma )}{\partial \gamma }=\frac{-4e^{\gamma }\left(1+me^{2\gamma }\right)}{k\left(1-\beta \right){\left(1-me^{2\gamma }\right)}^{2}}=0.$$

Since the term $e^{\gamma }$ is always positive, the value of the maximum, $\gamma_{max}$, can be obtained from the expression $\left(1+me^{2\gamma }\right)=0$, which results in
(53)$$\gamma_{max}=\frac{1}{2}ln\left(\frac{-1}{m}\right)\;m\leq 0.$$

It is shown that it is really a minimum because it verifies the second-order condition, i.e.,
(54)$$\frac{\partial^{2}f\left(\gamma \right)}{\partial \gamma^{2}}=\frac{4e^{\gamma }\left(1+me^{2\gamma }\left(6+me^{2\gamma }\right)\right)}{k\left(1-\beta \right){\left(-1+me^{2\gamma }\right)}^{3}},$$
replacing γ for this $\gamma_{max}$ of Equation (Equation 53), m=−0.1 and k=7, it can be verified that the result is positive:(55)$$\frac{\partial^{2}f\left(\gamma \right)}{\partial \gamma^{2}}=\frac{2e^{\gamma }}{k\left(1-\beta \right)}=\frac{0.897254}{1-\beta }>0\;\beta \leq 1.$$

We can observe the coincidence of the expressions of $\gamma_{0}$ where m>0 and $\gamma_{max}$ where m≤0, given by Equations (Equation 45) and (Equation 53), respectively. Therefore, we can write the following equation:(56)$$\gamma_{0}=\gamma_{max}=\frac{1}{2}ln\left(\frac{1}{\left|m\right|}\right)$$
Regarding the variation of the stability with γ, the behavior is different when $\alpha^{\prime }$ is negative and $\gamma <\gamma_{max}$, the stability zone is greater as γ increases in contrast to the case of $\alpha^{\prime }$ positive, where the stability zone decreases as γ increases. In this sense, considering the stability zone, the results obtained for m≤0 when $\alpha^{\prime }$ is negative are better than in the same case when $\alpha^{\prime }$ is positive.

As happens in the previous region considered, the bifurcation diagrams of output of the first player for the values γ=0 and γ=0.7 in the region m≤0, with m=−0.1 (we set $a=10,b=0.5,c_{1}=3,c_{2}=2.3$), are also represented in Figure 9 and Figure 10. In these figures, the increase of stability is shown for $\alpha^{\prime }<0$ when the value of β is greater, as well as for $\alpha^{\prime }>0$.

## 4. Profit

From the point of view of the quantum profit, we study the behavior of the benefit with and without memory with different values of the quantum entanglement (γ). We start from the quantum profit Equation (Equation 3), considering that there is not memory in t=0, then the quantum profit of the two firms can be obtained, replacing the initial value of both firms in Equation (Equation 3). After that, $x_{i}$ in t=1 is found from Equation (Equation 10), with $\mu_{i}\left(0\right)=x_{i}\left(0\right)$ being the initial value of the i-firm. With these values of $x_{i}\left(1\right)$, we can resolve the quantum profit of both firms in t=1, again with Equation (Equation 3). Repeating the process, we can find $x_{i}$ of both firms in t=2 with Equation (Equation 10), being $\mu_{i}\left(1\right)=x_{i}\left(1\right)$. With these values of $x_{i}\left(2\right)$, we can resolve the quantum profit in t=2 with Equation (Equation 3). For t=3, we can resolve the values $x_{i}\left(3\right)$ with Equation (Equation 10), calculating $\mu_{i}\left(2\right)$ from Equations (7)–(9). With these values, $\mu_{i}\left(2\right)$, the value of $x_{i}$ in t=3 can be resolved and so on to the next interaction of time.

### 4.1. Without Speed of Adjustment of the Boundedly Rational Player ($\alpha^{\prime }$)

When there is not speed of adjustment of the boundedly rational player ($\alpha^{\prime }$), the values of the quantum profits of both players are the same with or without memory. First of all, we represent the behavior of the quantum profit without memory of the two firms as a function of the quantum entanglement (γ) in the two cases, with m<0 and m>0. With m<0 ($c_{2}>c_{1}$, the cost of player 2 is greater than the cost of player 1), the quantum profit of the two firms with different (γ) can be seen in Figure 11, where the quantum profit in player 1 is lower than in player 2 when γ=0 (classic game). As the value of the degree of quantum entanglement (γ) increases, the quantum profits of both players move closer to the value of γ equal to 1.15, where both quantum profits are equal to 10.92. Due to the economic meaning of the game, this point is never reached because the value of γ has to be less than 1.09 according to Equation (Equation 47).

With m>0 ($c_{1}>c_{2}$, the cost of player 1 is greater than the cost of player 2), the quantum profit of the two firms with different (γ) is shown in Figure 12, where the quantum profit in player 1 is lower than in player 2 when γ=0 (classic game). If the value of the degree of quantum entanglement (γ) increases, the quantum profits of both players move closer but the profit of player 2 is always greater than the quantum profit of player 1. As in the previous case, when γ>1.15, the values have no economic meaning according to Equation (Equation 45).

### 4.2. With Positive Values of Speed of Adjustment of the Boundedly Rational Player ($\alpha^{\prime }$)

We study the behavior of the quantum profit with positive values of the speed of adjustment of the boundedly rational player ($\alpha^{\prime }$). We start with $\alpha^{\prime }=0.2$ and without memory in two cases, with m<0 and with m>0, representing them in Figure 13.

As is shown, if we remove the first 50 iterations, the quantum profit tends to be stable except when the value of γ is high, as can be seen in Figure 14.

In the system with memory (β=0.2), the chaotic effect observed in high values of γ is delayed, as can be seen in Figure 15.

When $\alpha^{\prime }$ increases, $x_{i}$ and, therefore, the quantum profit are outside the stability zone. Then, the profits can have different values in *t* and in t+1. The profits can vary between two or more values if the system is in the chaotic zone, as can be seen in Figure 16.

Considering the same value of $\alpha^{\prime }=0.4$ and applying a greater value of memory (β=0.5), we can see that the quantum profits do not vary between two or more values because the system is in the stability zone, as is represented in Figure 17.

### 4.3. With Negative Values of Speed of Adjustment of the Boundedly Rational Player ($\alpha^{\prime }$)

The next step is analyzing the behavior of the quantum profit with negative values of the speed of adjustment of the boundedly rational player ($\alpha^{\prime }$). Initially it is considered that $\alpha^{\prime }=-0.2$ and there is absence of memory in both cases, with m<0 and with m>0, as is shown in Figure 18.

As can be seen, the behavior is the same as if there is not speed of adjustment of the boundedly rational player, as in Figure 11 and Figure 12. That means that the effect of memory is the increase of the stability zone, delaying the presence of chaos.

Taking into account the Equation (Equation 50), in $\alpha^{\prime }<0$, when m<0 all values of γ are valid and therefore, there is a point in γ=1.15 where the quantum profits of both players have the same value and are equal to 10.91. However, when you increase γ, player 2 has more profit than the player 1. In m>0 only the values of γ<=1.15 are valid regarding to the Equation (Equation 45).

At this point, we increase the value of the speed of adjustment of the boundedly rational player to $\alpha^{\prime }=-0.8$ without applying memory. Then, the behavior is similar to the case with $\alpha^{\prime }=-0.2$ except in values of γ near 0, because the system is in the chaotic zone as can be seen in Figure 19. This is because it is outside of the stability zone, as is shown in Figure 2 and Figure 8.

If we add memory to the system, the stability zone increases and the profits do not vary chaotically, as happens in the example with α=−0.8 and β=0.8, which can be seen in Figure 20.

## 5. Conclusions

In this paper, we analyzed the effect of memory in a dynamical Cournot duopoly game with heterogeneous players (one of them is boundedly rational and the other one, a naive player), applying a quantum scheme when the quantum entanglement (γ) has values greater than one and the speed of adjustment ($\alpha^{\prime }$) is negative. We consider these special and unusual values of the parameters to study the behavior of local stability and profit in this case, in contrast to the conventional values. In all this analysis, we only took into consideration the cases with economic meaning, obtaining the values of the parameters where the study is valid. Additionally, to analytically verify the statements, we represented them graphically and we performed some numerical simulations to visualizethe results obtained.

Considering a negative speed of adjustment, it is proved that the stability zone in the game with memory is greater than in the classic game and it also increases when the memory factor is greater, as happens with a positive speed of adjustment. It verifies that the stability is greater in the negative than in the positive zone of the speed of adjustment, improving, in this sense, the results previously obtained.

Regarding the variation of the stability with γ, when the relative marginal cost *m* is positive and $\alpha^{\prime }$ is negative, the stability zone increases if γ increases. This is the same behavior previously observed when $\alpha^{\prime }$ is positive, except in the interval of *m* between 0 and 0.1, where the stability decreases as γ increases. Considering the case where m≤0 and $\alpha^{\prime }$ is negative, the stability zone is greater as γ increases. This is the opposite behavior to the case of positive values of $\alpha^{\prime }$, where the stability decreases as γ increases. This increase of stability when $\alpha^{\prime }<0$ enables the model to reach Nash equilibrium faster as the speed of adjustment increases, reaching the equilibrium payoff in fewer steps, compared to the case of $\alpha^{\prime }>0$.

In the analysis of the profit, when $\alpha^{\prime }=0$, it is shown that the memory has no influence on the results and it is observed that the profit of both players tends to move closer, but there is not a point of coincidence of both profits in the zone of economic meaning. In the cases with $\alpha^{\prime }>0$ and $\alpha^{\prime }<0$, the memory influences the results in the sense of causing a delay in the dynamics.

In summary, it can be considered that the results obtained with speed of adjustment less than zero and degree of entanglement greater than one in some of the analyzed cases improved the previous ones, which shows progress in this area. As we mentioned, we focused on the economic field, but we hope this study can also be useful in other fields.

## Figures and Tables

**Figure 1 entropy-24-01333-f001:**
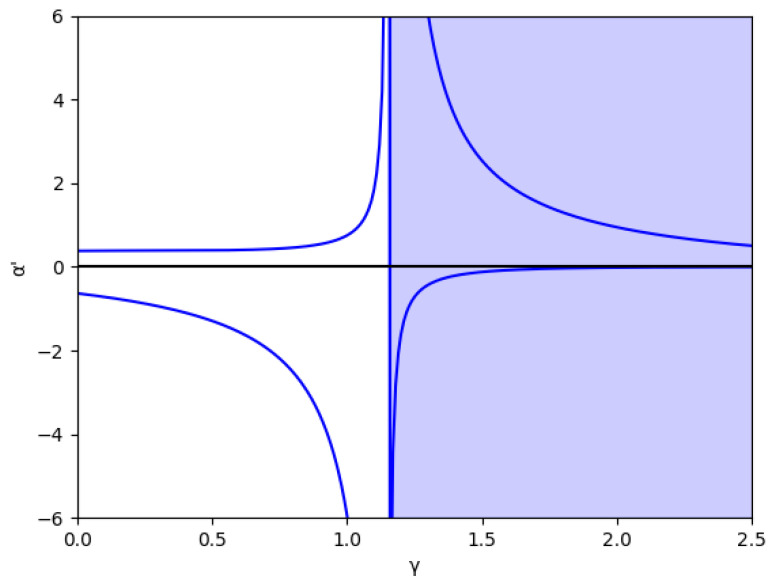
Stability region for m=0.1 ($a=10,\;b=0.5,\;c_{1}=3,\;c_{2}=2.3$) in the absence of memory. The area with no economic sense is shaded in blue, that is, when $\gamma >\gamma_{0}=1.1513$.

**Figure 2 entropy-24-01333-f002:**
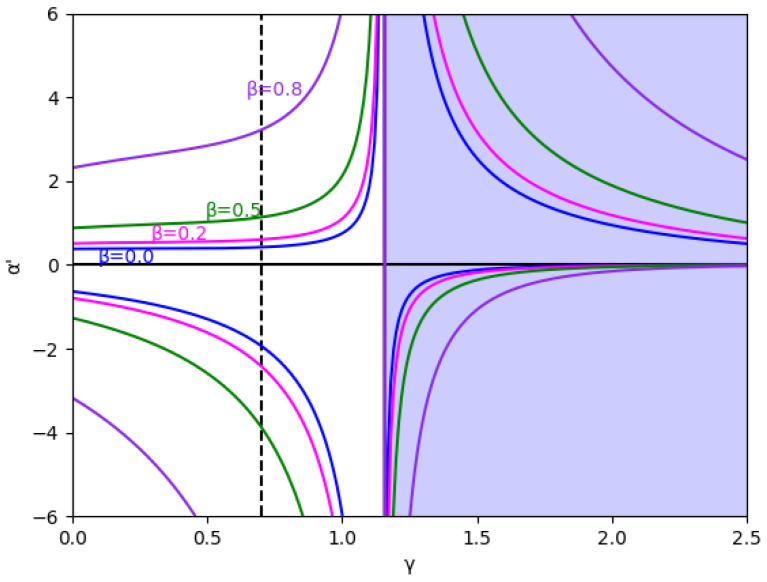
Stability region for m=0.1 ($a=10,\;b=0.5,\;c_{1}=3,\;c_{2}=2.3$) and different values of β (0, 0.2, 0.5, and 0.8). The vertical line γ=0.7 is represented because this value and γ=0 are the values used in the bifurcation diagrams. The area with no economic sense is shaded in blue, that is, when $\gamma >\gamma_{0}=1.1513$.

**Figure 3 entropy-24-01333-f003:**
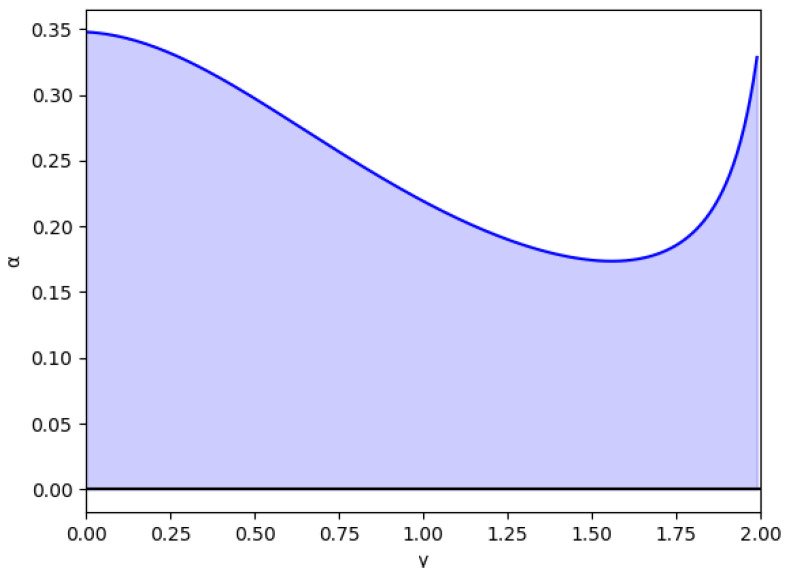
Stability region as a function of γ for m=0.014 ($a=10,\;b=0.5,\;c_{1}=3,\;c_{2}=2.9$) and without memory, β=0.

**Figure 4 entropy-24-01333-f004:**
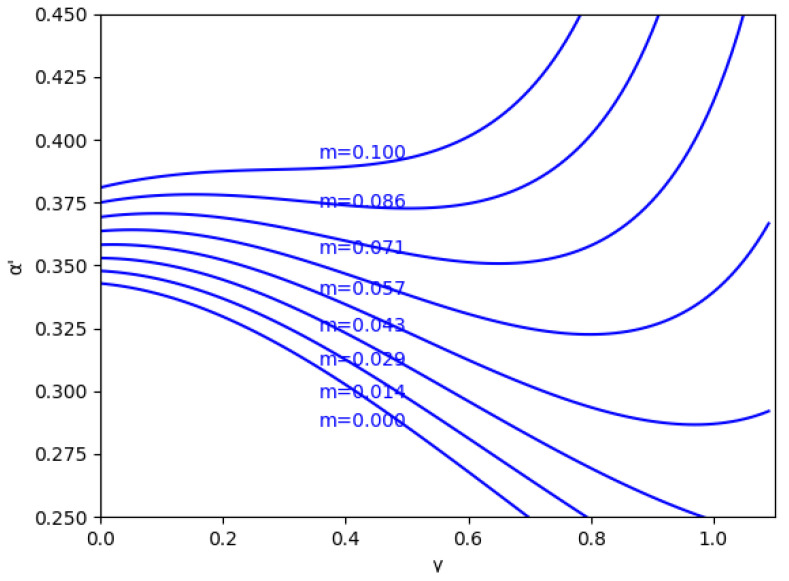
Stability region as a function of γ for *m* between 0.009<m<0.1 and without memory, β=0.

**Figure 5 entropy-24-01333-f005:**
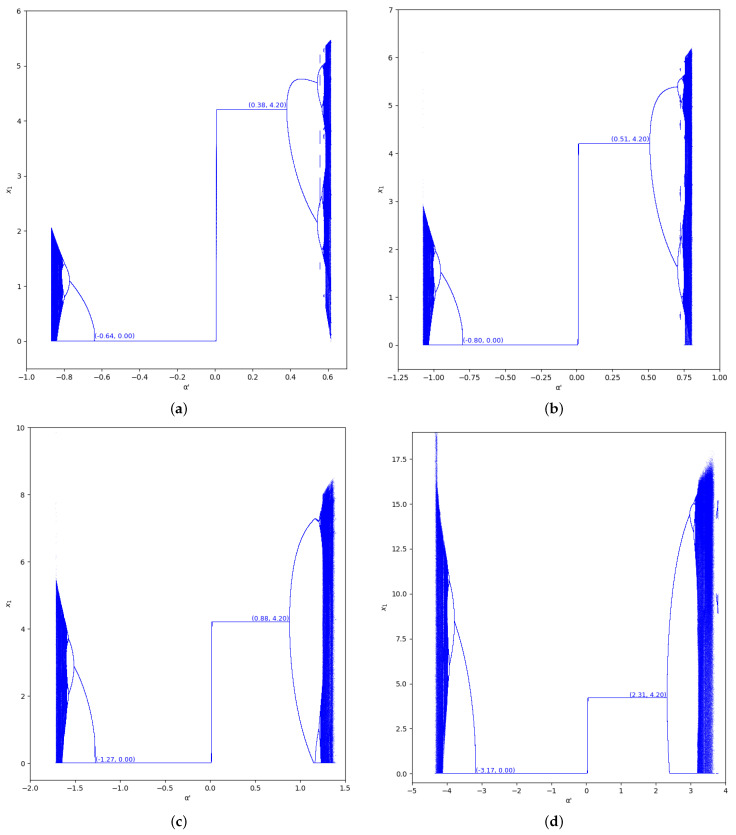
Bifurcation diagrams of output of the first player as a function of α with γ=0 and β=0,0.2,0.5,0.8 in the zone m=0.1 ($a=10,b=0.5,c_{1}=3,c_{2}=2.3$). (**a**) β=0; (**b**) β=0.2; (**c**) β=0.5; (**d**) β=0.8.

**Figure 6 entropy-24-01333-f006:**
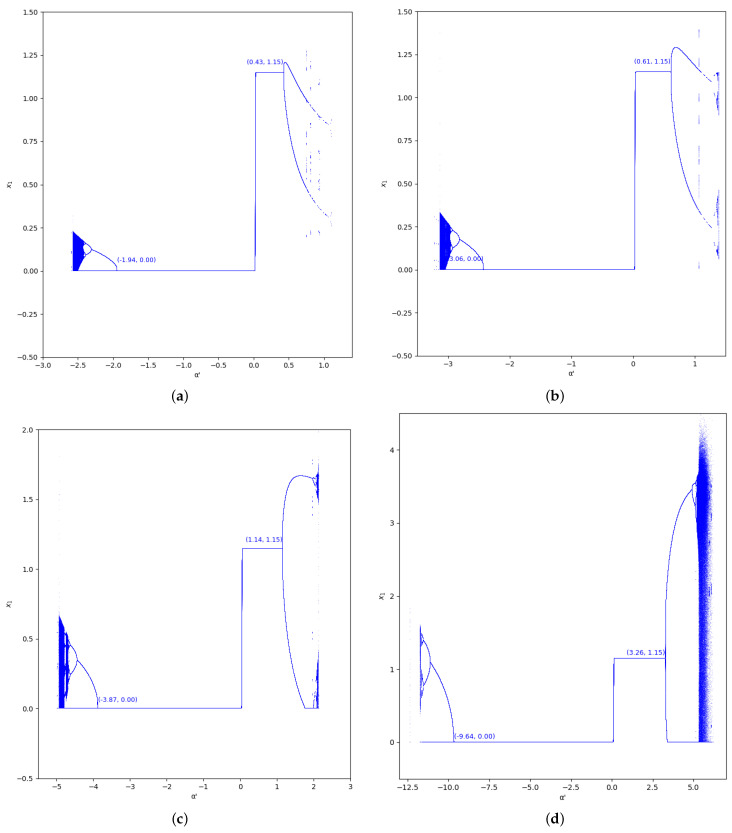
Bifurcation diagrams of output of the first player as a function of α with γ=0.7 and β=0,0.2,0.5,0.8 in the zone m=0.1 ($a=10,b=0.5,c_{1}=3,c_{2}=2.3$). (**a**) β=0; (**b**) β=0.2; (**c**) β=0.5; (**d**) β=0.8.

**Figure 7 entropy-24-01333-f007:**
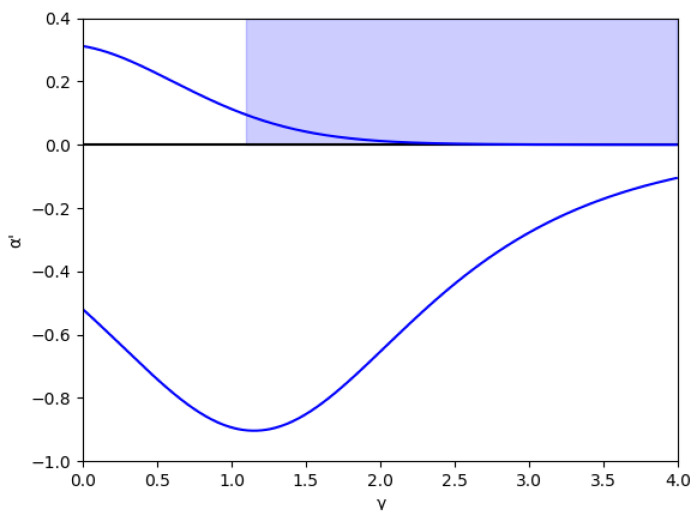
Stability region for m=−0.1 ($a=10,\;b=0.5,\;c_{1}=3,\;c_{2}=3.7$) in the absence of memory. The area with no economic sense is shaded in blue, that is, when $\gamma >\gamma_{1}=1.0986$ and $\alpha^{\prime }>0$.

**Figure 8 entropy-24-01333-f008:**
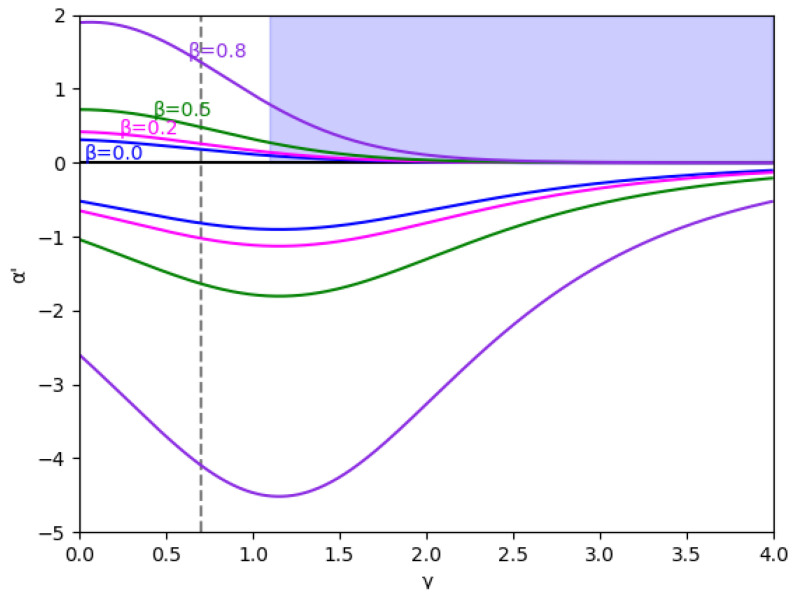
Stability region for m=−0.1 ($a=10,\;b=0.5,\;c_{1}=3,\;c_{2}=3.7$) and different values of β (0, 0.2, 0.5, and 0.8). The vertical line γ=0.7 is represented because this value and γ=0 are the values used in the bifurcation diagrams. The area with no economic sense is shaded in blue, that is, when $\gamma >\gamma_{1}=1.0986$ and $\alpha^{\prime }>0$.

**Figure 9 entropy-24-01333-f009:**
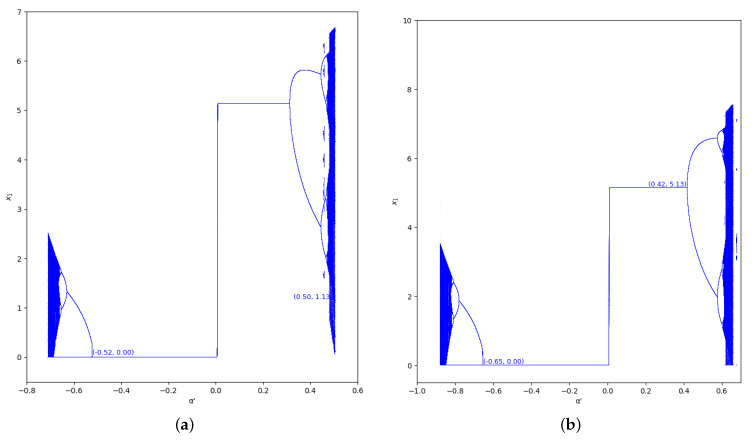
Bifurcation diagrams of output of the first player as a function of α with γ=0 and β=0, 0.2, 0.5, 0.8 in the zone m=−0.1 ($a=10,\;b=0.5,\;c_{1}=3,\;c_{2}=3.7$). (**a**) β=0; (**b**) β=0.2; (**c**) β=0.5; (**d**) β=0.8.

**Figure 10 entropy-24-01333-f010:**
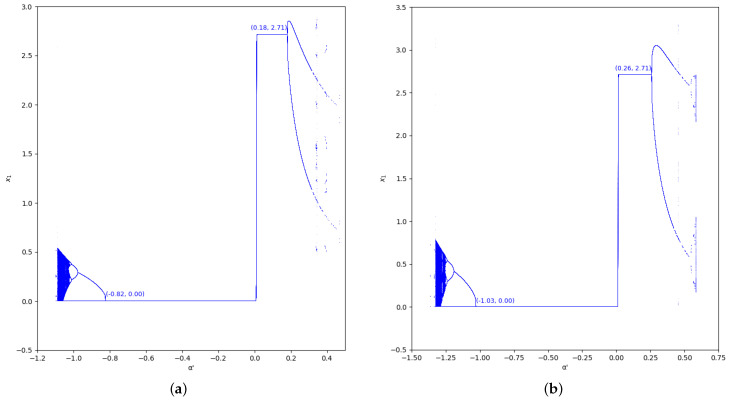
Bifurcation diagrams of output of the first player as a function of α with γ=0.7 and β=0, 0.2, 0.5, 0.8 in the zone m=−0.1 ($a=10,\;b=0.5,\;c_{1}=3,\;c_{2}=3.7$). (**a**) β=0; (**b**) β=0.2; (**c**) β=0.5; (**d**) β=0.8.

**Figure 11 entropy-24-01333-f011:**
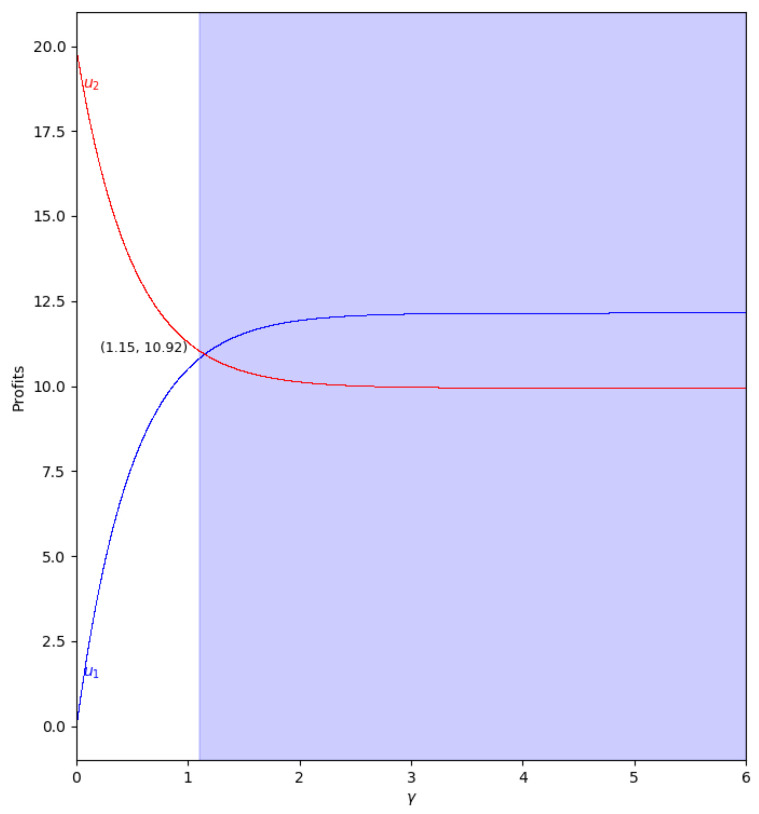
Quantum profits of the two firms ($u_{1}$ in blue and $u_{2}$ in red) as a function of γ (degree of quantum entanglement) with β=0 (without memory) and $\alpha^{\prime }$=0 (speed of adjustment is zero, therefore $x_{1}\left(t+1\right)=x_{1}\left(t\right)$) in the zone m=−0.1 ($a=10,\;b=0.5,\;c_{1}=3,\;c_{2}=3.7$) with initial value of $x_{1}\left(0\right)=x_{2}\left(0\right)=0.001$. The point where $u_{1}=u_{2}=10.92$ is when γ=1.15, but according to Equation (Equation 47), γ≤1.09 to have economic meaning.

**Figure 12 entropy-24-01333-f012:**
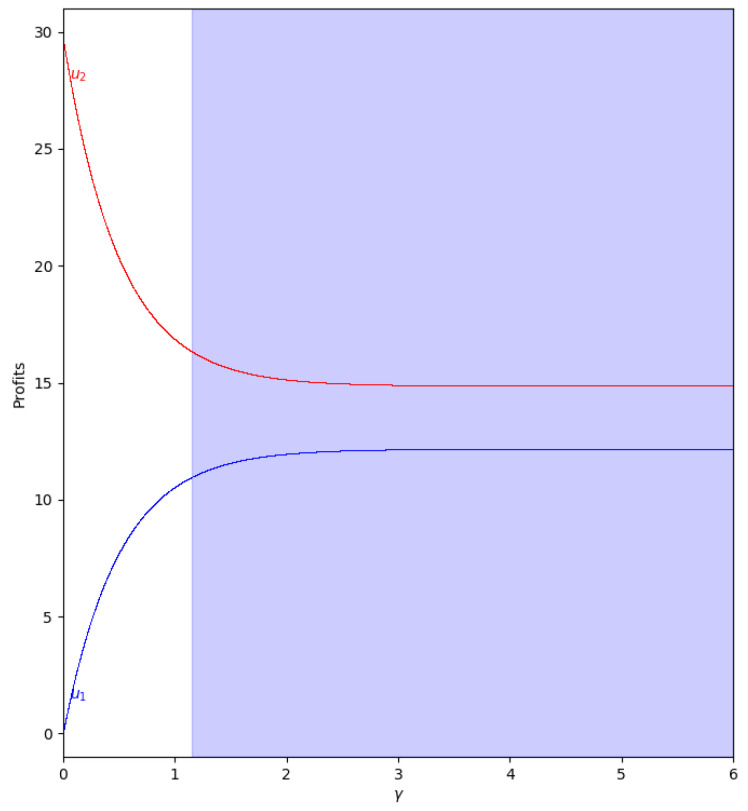
Quantum profits of the two firms ($u_{1}$ in blue and $u_{2}$ in red) as a function of γ (degree of quantum entanglement) with β=0 (without memory) and $\alpha^{\prime }$=0 (speed of adjustment is zero, therefore $x_{1}\left(t+1\right)=x_{1}\left(t\right)$) in the zone m=0.1 ($a=10,\;b=0.5,\;c_{1}=3,\;c_{2}=2.3$) with initial value of $x_{1}\left(0\right)=x_{2}\left(0\right)=0.001$. When γ≤1.15 the game has economic meaning according to Equation (Equation 45).

**Figure 13 entropy-24-01333-f013:**
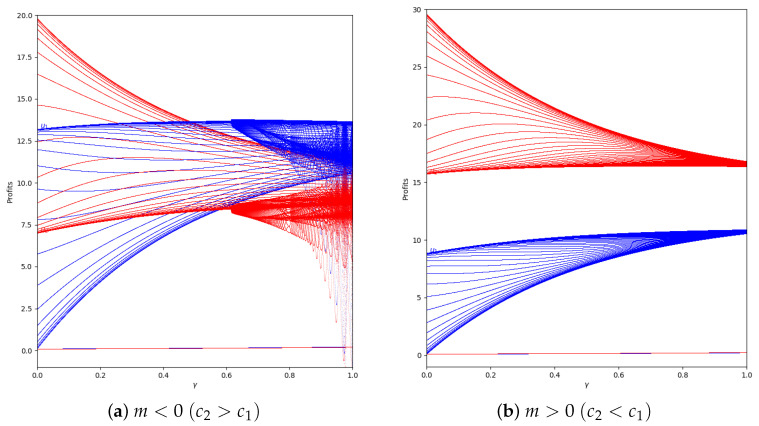
Quantum profits of the two firms ($u_{1}$ in blue and $u_{2}$ in red) as a function of γ (degree of quantum entanglement) with β=0 (without memory) and $\alpha^{\prime }$ = 0.2 (speed of adjustment): (**a**) in the zone m=−0.1 (a=10, b=0.5, $c_{1}=3,\;c_{2}=3.7$) and (**b**) in the zone m=0.1 (a=10, b=0.5, $c_{1}=3,\;c_{2}=2.3$), with initial value of $x_{1}\left(0\right)=x_{2}\left(0\right)=0.001$.

**Figure 14 entropy-24-01333-f014:**
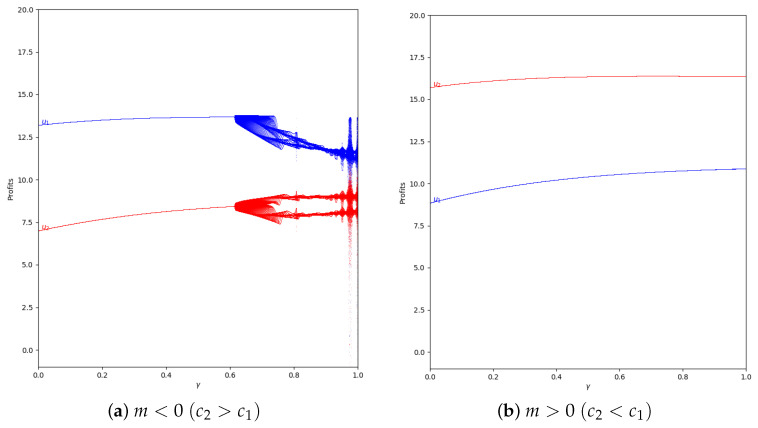
Quantum profits of the two firms ($u_{1}$ in blue and $u_{2}$ in red) as a function of γ (degree of quantum entanglement) with β=0 (without memory), $\alpha^{\prime }$ = 0.2 (speed of adjustment), and without the first 50 iterations: (**a**) in the zone m=−0.1 ($a=10,\;b=0.5,\;c_{1}=3,\;c_{2}=3.7$) and (**b**) in the zone m=0.1 ($a=10,\;b=0.5,\;c_{1}=3,\;c_{2}=2.3$), with initial value of $x_{1}\left(0\right)=x_{2}\left(0\right)=0.001$.

**Figure 15 entropy-24-01333-f015:**
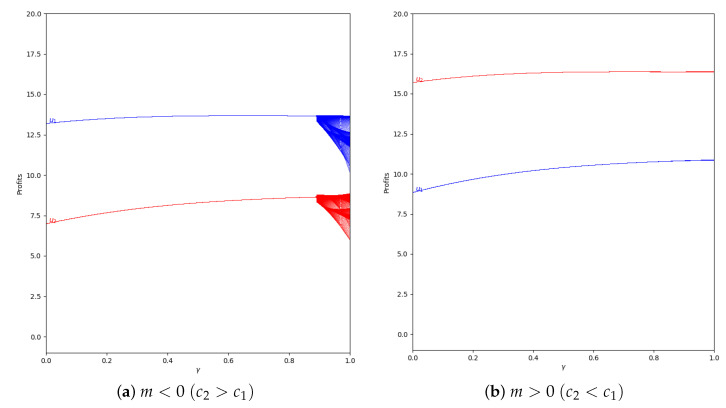
Quantum profits of the two firms ($u_{1}$ in blue and $u_{2}$ in red) as a function of γ (degree of quantum entanglement) with β=0.2 (with memory), $\alpha^{\prime }$ = 0.2 (speed of adjustment), and without the first 50 iterations: (**a**) in the zone m=−0.1 ($a=10,\;b=0.5,\;c_{1}=3,\;c_{2}=3.7$) and (**b**) m=0.1 ($a=10,\;b=0.5,\;c_{1}=3,\;c_{2}=2.3$), with initial value of $x_{1}\left(0\right)=x_{2}\left(0\right)=0.001$.

**Figure 16 entropy-24-01333-f016:**
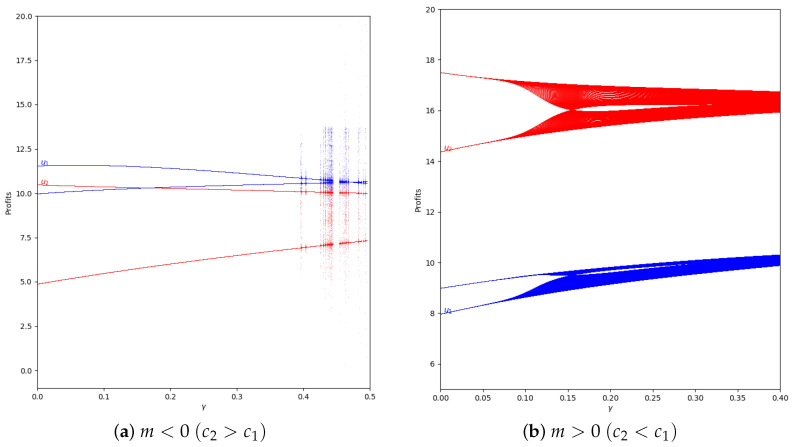
Quantum profits of the two firms ($u_{1}$ in blue and $u_{2}$ in red) as a function of γ (degree of quantum entanglement) with β=0 (without memory), $\alpha^{\prime }$ = 0.4 (speed of adjustment), and without the first 50 iterations: (**a**) in the zone m=−0.1 ($a=10,\;b=0.5,\;c_{1}=3,\;c_{2}=3.7$) and (**b**) m=0.1 ($a=10,\;b=0.5,\;c_{1}=3,\;c_{2}=2.3$), with initial value of $x_{1}\left(0\right)=x_{2}\left(0\right)=0.001$.

**Figure 17 entropy-24-01333-f017:**
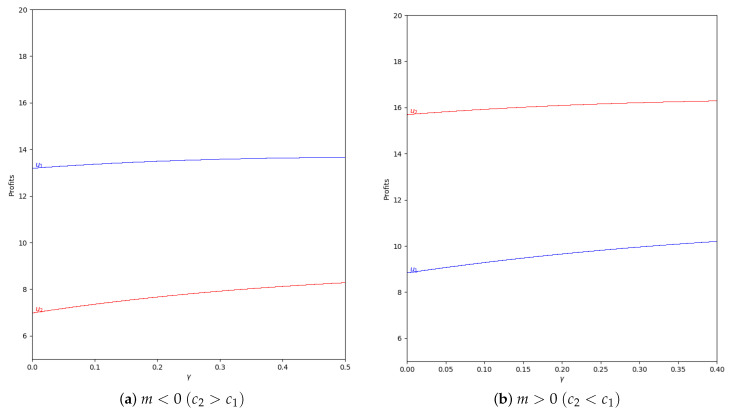
Quantum profits of the two firms ($u_{1}$ in blue and $u_{2}$ in red) as a function of γ (degree of quantum entanglement) with β=0.5 (with memory), $\alpha^{\prime }$ = 0.4 (speed of adjustment), and without the first 100 iterations: (**a**) in the zone m=−0.1 ($a=10,\;b=0.5,\;c_{1}=3,\;c_{2}=3.7$) and (**b**) m=0.1 ($a=10,\;b=0.5,\;c_{1}=3,\;c_{2}=2.3$), with initial value of $x_{1}\left(0\right)=x_{2}\left(0\right)=0.001$.

**Figure 18 entropy-24-01333-f018:**
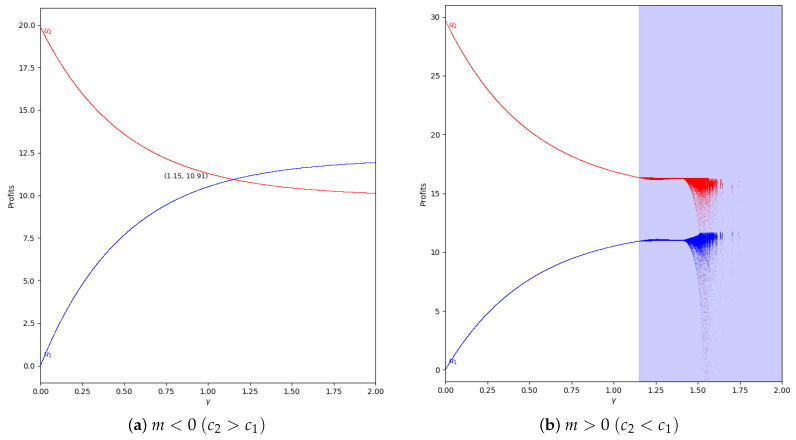
Quantum profits of the two firms ($u_{1}$ in blue and $u_{2}$ in red) as a function of γ (degree of quantum entanglement) with β=0 (without memory) and $\alpha^{\prime }$ = −0.2 (speed of adjustment): (**a**) in the zone m=−0.1 ($a=10,\;b=0.5,\;c_{1}=3,\;c_{2}=3.7$) and (**b**) in the zone m=0.1 (a=10, b=0.5, $c_{1}=3,\;c_{2}=2.3$), with initial value of $x_{1}\left(0\right)=x_{2}\left(0\right)=0.001$. Regarding Equation (Equation 50), in $\alpha^{\prime }<0$, when m<0 all values of γ are valid, but when m>0, only the values of γ≤1.15 are valid according to Equation (Equation 45).

**Figure 19 entropy-24-01333-f019:**
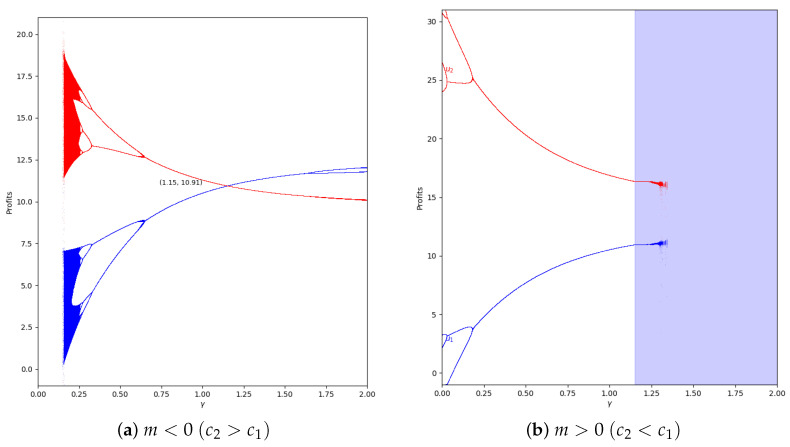
Quantum profits of the two firms ($u_{1}$ in blue and $u_{2}$ in red) as a function of γ (degree of quantum entanglement) with β=0 (without memory) and $\alpha^{\prime }$ = −0.8 (speed of adjustment): (**a**) in the zone m=−0.1 ($a=10,\;b=0.5,\;c_{1}=3,\;c_{2}=3.7$) and (**b**) in the zone m=0.1 (a=10, b=0.5, $c_{1}=3,\;c_{2}=2.3$), with initial value of $x_{1}\left(0\right)=x_{2}\left(0\right)=0.001$. Regarding Equation (Equation 50), in $\alpha^{\prime }<0$, when m<0 all values of γ are valid, but when m>0, only the values of γ≤1.15 are valid according to Equation (Equation 45).

**Figure 20 entropy-24-01333-f020:**
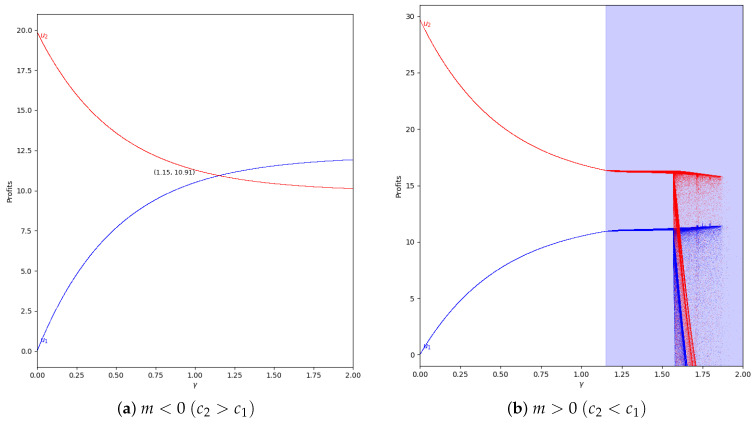
Quantum profits of the two firms ($u_{1}$ in blue and $u_{2}$ in red) as a function of γ (degree of quantum entanglement) with β=0.8 and $\alpha^{\prime }=-0.8$ (speed of adjustment): (**a**) in the zone m=−0.1 ($a=10,\;b=0.5,\;c_{1}=3,\;c_{2}=3.7$) and (**b**) in the zone m=0.1 ($a=10,b=0.5,c_{1}=3,c_{2}=2.3$), with initial value of $x_{1}\left(0\right)=x_{2}\left(0\right)=0.001$. Regarding Equation (Equation 50), in $\alpha^{\prime }<0$, when m<0, all values of γ are valid, but when m>0, only the values of γ≤1.15 are valid according to Equation (Equation 45).

## Data Availability

The data that support the findings of this study are available within the article.

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
