# Peer review of "Complex Dynamics of a Cournot Quantum Duopoly Game with Memory and Heterogeneous Players"

_entropy, 2022, doi:10.3390/e24101333_

Round 1
Reviewer 1 Report
The paper analyses the quantum Cournot model under "minimal" quantization rules from Li-Du-Massar. This means effectively that prices $q_1,q_2$, which are strategic variables in the classical model, are now obtained by Lorentz rotation from the new strategic variables $x_1,x_2$. Thus all quantum content is captured by a rotation of strategic variables. The dynamics of adjustments is introduced for heterogeneous players as in the paper of Lian-Feng. Then the dynamics is made more complicated by introducing additional parameters responsible for memory, in the way suggested in a previous paper of the authors. The fixed points of the resulting discrete dynamics on $\R^2$ are found explicitly and their stability is analysed via the standard calculations with the Jacobian. Special efforts are devoted to the values of parameters that have no economic meaning (negative prices!)
I think the contribution is not very essential for a sound publication.
Moreover, the paper is presented rather carelessly.
Examples:
The first sentence of the introduction (and of abstract) is not clear at all.
Lines 49,50 bad English making again the meaning not clear.
F-la (2). Meant to be the positive part of this number, isn't it?
F-la (4): What is tialu? or tialx? Moreover these formulas are given without any explanation at all. In fact, they are just copied from [11], which should be clearly stated.
From f-la (6) to the end of subsection there is complete mess with the notations. Symbol $t$ jumps randomly from lower index to a place in the bracket, and also randomly becomes capital (like $\mu_T$)! Thus the main parameters are not even properly introduced! Moreover, if $\mu_1=x_1$ (as seemingly is the case, line 96), then why introduce $\mu_1$ at all? And in f-la (13) it states that $\mu_i(t)=x_i(t)$!?
The fixed points of dynamics under study are suddenly called Nash equilibria (after f-la (17)) without even an attempt to state for which games they can be referred.
What is the point to investigate negative $\alpha$ that have no economic meaning (negative prices?). I see no sufficient motivations for presenting rather trivial calculations with 2 times 2 matrices.
For all these reasons I would rather suggest to reject this contribution.
Author Response
The paper analyses the quantum Cournot model under "minimal" quantization rules from Li-Du-Massar. This means effectively that prices $q_1,q_2$, which are strategic variables in the classical model, are now obtained by Lorentz rotation from the new strategic variables $x_1,x_2$. Thus all quantum content is captured by a rotation of strategic variables. The dynamics of adjustments is introduced for heterogeneous players as in the paper of Lian-Feng. Then the dynamics is made more complicated by introducing additional parameters responsible for memory, in the way suggested in a previous paper of the authors. The fixed points of the resulting discrete dynamics on $\R^2$ are found explicitly and their stability is analysed via the standard calculations with the Jacobian. Special efforts are devoted to the values of parameters that have no economic meaning (negative prices!)
>> Quantity, and not price, is the main variable in the Cournot duopoly but price is present in the analysis. To clarify that we don’t use negative prices, we have added some text after Eq. (2) where it is explained that the demand ‘a’ is greater than ‘bQc’ in that equation. If you are referring to negative quantities, we don’t consider negative values, as you can check in the figures 5, 6, 9 and 10 and also with the comment in the paragraph before the subsection 2.1. Related to the economic meaning, when the speed of adjustment is negative it means that the production of a year is less than in the previous one, which is a case that happens frequently in the real world. At last, as it is analyzed in the article, some values of the degree of entanglement greater than one are possible maintaining the economic meaning.
I think the contribution is not very essential for a sound publication.
Moreover, the paper is presented rather carelessly.
Examples:
The first sentence of the introduction (and of abstract) is not clear at all.
>> First part of introduction and abstract have been rewritten to be better understood
Lines 49,50 bad English making again the meaning not clear.
>> This part has been rewritten to be better understood.
F-la (2). Meant to be the positive part of this number, isn't it?
>> It has been rewritten to be better understood, clarifying that it is positive because a > bQc.
F-la (4): What is tialu? or tialx? Moreover these formulas are given without any explanation at all. In fact, they are just copied from [11], which should be clearly stated.
>> Both of them, tialu and tialx, are mistakes related to the change of template. We have corrected them and some text has added in section 2 to explain more clearly how the formulas are obtained.
From f-la (6) to the end of susection there is complete mess with the notations. Symbol $t$ jumps randomly from lower index to a place in the bracket, and also randomly becomes capital (like $\mu_T$)! Thus the main parameters are not even properly introduced! Moreover, if $\mu_1=x_1$ (as seemingly is the case, line 96), then why introduce $\mu_1$ at all? And in f-la (13) it states that $\mu_i(t)=x_i(t)$!?
>> We agree that the notation wasn’t correct at all. We have reviewed it to improve the readability and comprehension.
The fixed points of dynamics under study are suddenly called Nash equilibria (after f-la (17)) without even an attempt to state for which games they can be referred.
>> The existence of Nash equilibrium in games is closely related to some theorems such as the Brouwer and Kakutani’s fixed point theorems. Indeed, John Nash used these theorems to proof the existence of Nash equilibrium under conditions equivalent to our study. Due to this point, fixed points are considered as Nash equilibria. These results are well known and there are a lot of examples of similar analysis to our one in the bibliography.
What is the point to investigate negative $\alpha$ that have no economic meaning (negative prices?). I see no sufficient motivations for presenting rather trivial calculations with 2 times 2 matrices.
>> As we have previously mentioned, a negative speed of adjustment has economic meaning because it indicates that the production of a year is less than in the previous one. Then, if the rest of parameters don’t vary, the profit in the current year will be lower than in the previous one.
For all these reasons I would rather suggest to reject this contribution.
Reviewer 2 Report
The authors addressed a quantum model of Cournot competition.
They present (classical) chaotic properties of the solutions when there is entanglement.
However this is very hard to read. The model of this study is poorly explained and it is not at all clear under what circumstances this game might be played. Furthermore, the explanation regarding the motivation and background for this study is insufficient, so a clear explanation that readers can understand is required.
So I would like to suggest revisions as follows.
1) What situations are envisioned for the creation of quantum informational commodities as intended in this study? For example, the following paper details the relationship between quantum commodities and quantum money. https://link.springer.com/article/10.1007/s11128-021-03378-5
2) Is there any quantum supremacy in the model being presented? The presence of quantum entanglement does not enough, as it can be efficiently realized by classical computers. For this point, to my best knowledge, the only papers which show supremacy are https://link.springer.com/article/10.1007/s11128-021-03252-4 and the reference in previous comment 1). The authors need to clarify what makes this game more interesting compared with classical ones.
3) The figures must be improved. Almost all of the figures are merely plots of numerical results and do not seem to have any significant content.
4) Mathematical/Theoretical presentation of the work is poor. They are presenting a number of equations, but not formally prepared. Since the form used in this model is essentially an EWL method, a general equilibrium solution may be expressed. To improve this, see
https://arxiv.org/abs/1801.02053 and https://arxiv.org/abs/1803.07919
5) The authors need to make more general claims about the development of the study.
For example, the Cournot competition model can be extended to repeated games, and that is more natural in actual economics. In the course of long-term competition in the market, contracts may need to be renegotiated, and in such cases, it will be possible to develop interesting arguments from the perspective of incomplete contract theory. https://link.springer.com/article/10.1007/s11128-019-2519-8 https://link.springer.com/article/10.1007/s11128-021-03295-7
Author Response
The authors addressed a quantum model of Cournot competition.
They present (classical) chaotic properties of the solutions when there is entanglement.
However this is very hard to read. The model of this study is poorly explained and it is not at all clear under what circumstances this game might be played. Furthermore, the explanation regarding the motivation and background for this study is insufficient, so a clear explanation that readers can understand is required.
So I would like to suggest revisions as follows.
1) What situations are envisioned for the creation of quantum informational commodities as intended in this study? For example, the following paper details the relationship between quantum commodities and quantum money. https://link.springer.com/article/10.1007/s11128-021-03378-5
>> The cited article analyzes quantum games from a different point of view. This paper describes economic activities from a viewpoint of quantum games. In the market it is defined a quantum commodity as a good which can be interchangeable between players to trade and it is studied how some of these goods, called quantum commodity money, can emerge as a media of exchange not to be consumed or used in production, under certain conditions. In our case, on the basis of the classic Cournot duopoly, the resulting quantities have been quantized following the Li-Du-Massar scheme [7] with the purpose of obtaining better results than in the classic games, as other authors have previously done successfully. This is the only objective of including quantization in the game model. Anyway, as we considered it interesting, we have cited the proposed article https://link.springer.com/article/10.1007/s11128-021-03378-5.
2) Is there any quantum supremacy in the model being presented? The presence of quantum entanglement does not enough, as it can be efficiently realized by classical computers. For this point, to my best knowledge, the only papers which show supremacy are https://link.springer.com/article/10.1007/s11128-021-03252-4 and the reference in previous comment 1). The authors need to clarify what makes this game more interesting compared with classical ones.
>> As the reference in previous comment, this paper tackles the quantum issue from a different view. In our article, we apply quantum game theory to include entanglement in the independent quantities of the Cournot duopoly game with memory, obtaining better results than in the classic game. Quantum game theory is not considered in the proposed article because it seems that the author is not so in favour of it. Indeed, there is an extract in proposed article where he explains it: “Therefore, in this paper, we will not adopt methods such as quantum game theory as done in the past, but will choose to extend conventional microeconomic models in terms of quantum theory”.
3) The figures must be improved. Almost all of the figures are merely plots of numerical results and do not seem to have any significant content.
>>Effectively, the main purpose of the figures is to support the results of the previous equations. They are used to see more clearly the increase in stability with memory and the status of the payoffs.
4) Mathematical/Theoretical presentation of the work is poor. They are presenting a number of equations, but not formally prepared. Since the form used in this model is essentially an EWL method, a general equilibrium solution may be expressed. To improve this, see
https://arxiv.org/abs/1801.02053 and https://arxiv.org/abs/1803.07919
>>It is correct, we apply a model of quantization but there are other examples of similar models, such as EWL or MW schemes. Specifically, as you comment, Eisert et al. described the EWL model in their paper published in 1999 (https://arxiv.org/abs/quant-ph/9806088) and applied it to the Prisoners’ Dilemma, showing that this game ceases to pose a dilemma if quantum strategies are allowed for. Some years later, in 2003, Li et al. explained their model of quantization, the LDM scheme, and, for the case of Cournot duopoly game, they showed that when the entanglement increases, the profits increases. We have used this model of quantization since it has been applied succesfully by other authors not only in Cournot duopoly game [8] [11], but also in Bertrand duopoly game [9] and Stackelberg duopoly game [11]. As you suggest, we have made some changes to improve the description of the LDM in section 2 to clarify some aspects. A full and detailed description of this method of quantization is explained in [7], since it is not the purpose of our paper to focus on it. Anyway, since we considered them relevant, we have cited one of the articles you suggest, https://arxiv.org/abs/1803.07919, and also the paper where it is described the EWL quantization scheme, https://arxiv.org/abs/quant-ph/9806088.
5) The authors need to make more general claims about the development of the study.
For example, the Cournot competition model can be extended to repeated games, and that is more natural in actual economics. In the course of long-term competition in the market, contracts may need to be renegotiated, and in such cases, it will be possible to develop interesting arguments from the perspective of incomplete contract theory. https://link.springer.com/article/10.1007/s11128-019-2519-8 https://link.springer.com/article/10.1007/s11128-021-03295-7
>> In future articles we will address it, since your suggestions are interesting ways to continue studying the Cournot duopoly game.
Reviewer 3 Report
It is hard to assess the value added by the manuscript. The provided list of reference is incomplete, and the paper does not include a proper discussion of the state-of-the-art. The results are presented in a chaotic manner, with low-quality, hard to read plots, and unclear formulas. Authors should significantly improve the quality of presentation. Moreover, no collusions are provided and no relation to the previous work is included. These points make the paper unsuitable for publication in the current form.
Author Response
It is hard to assess the value added by the manuscript. The provided list of reference is incomplete, and the paper does not include a proper discussion of the state-of-the-art. The results are presented in a chaotic manner, with low-quality, hard to read plots, and unclear formulas. Authors should significantly improve the quality of presentation. Moreover, no collusions are provided and no relation to the previous work is included. These points make the paper unsuitable for publication in the current form.
>> Some parts of the article have been rewritten to be better understood, taking into account your suggestions. For example, we have added some explanations to improve the quality of the analitical part and some sections have been modified to facilitate the comprehension, i.e. the abstract and the introduction.
Reviewer 4 Report
The manuscript requires a severe linguistic correction, which may remove some of the caveats cited below.
In general, the current version of the manuscript is incomprehensible in many places:
1. The authors claim that they are considering a quantum game. I think the only connection with quantum games is using the payoff function given by equation (1) quoted from [7]. With this approach, we are dealing with a classic game with a payout function (1). I do not know how the authors implement the memory and heterogeneity of players in a quantum game. This issue must be clarified.
2. What do the authors understand by the term "fixed point"?
The context shows that they mean solutions to a system of algebraic equations rather than fixed points of the mapping.
What is "the speed of adjustment"?
3. Part three is entitled "Numerical simulations," but I only find graphs of functions. What exactly and how is it simulated?
4. Authors must correct the manuscript to make it understandable. The beginnings of the Abstract and Introduction are pure nonsense!
Without clarifying the above doubts, I cannot recommend the publication of the manuscript.
Author Response
The manuscript requires a severe linguistic correction, which may remove some of the caveats cited below.
In general, the current version of the manuscript is incomprehensible in many places:
- The authors claim that they are considering a quantum game. I think the only connection with quantum games is using the payoff function given by equation (1) quoted from [7]. With this approach, we are dealing with a classic game with a payout function (1). I do not know how the authors implement the memory and heterogeneity of players in a quantum game. This issue must be clarified.
>> Firstly, some changes have been introduced in section 2 to correct some mistakes and improve the quality of the text, specially in the part related to the hetereogeneity of players and implementation of the memory. Regarding to the quantization of the game, we have only quantized the quantities of the two firms. As the price depends on the quantities and the payoff function depends on the quantities and the prices, then both parameters, prices and payoffs, are also quantized. Finally, we have included memory in the model in the sense that the quantities of the firms at one step depend on the quantities in previous steps, not only in the last step.
- What do the authors understand by the term "fixed point"?
The context shows that they mean solutions to a system of algebraic equations rather than fixed points of the mapping.
>> A fixed point is a value that does not change under a given transformation. Specifically, in mathematics, a fixed point of a function is an element that is mapped to itself by the function. In the case of the paper, we have analyzed a dynamical system where we use iterations to know the behaviour of this system in the differents steps of time. If the system converge to a point when we iterate it, then we have a fixed point. Under conditions equivalent to the considered in the article, John Nash showed that fixed points represent Nash equilibria in this case.
What is "the speed of adjustment"?
>> Since the knowledge of the market is usually incomplete, the first firm with boundedly rational expectations builds his output decision on the basis of the estimation of the marginal profit ∂u1(t)/∂x1(t) . This player decides to increase or decrease the production if the marginal profit is positive or negative, respectively. This mechanism where a firm adjusts his output according to the margin profit has been called myopic and is adjusted by the speed of adjustment.
- Part three is entitled "Numerical simulations," but I only find graphs of functions. What exactly and how is it simulated?
>> It is a numerical simulation because it is a calculation which is run on a computer following a program that implements in Python a mathematical model for the dynamical system studied in the article. The results must be consistent with the analytical solution previously obtained, as it really occurs.
- Authors must correct the manuscript to make it understandable. The beginnings of the Abstract and Introduction are pure nonsense!
>> The beginnings of the Abstract and Introduction have been rewritten to be better understood.
Without clarifying the above doubts, I cannot recommend the publication of the manuscript.
Round 2
Reviewer 1 Report
The text is improved after revision
Reviewer 2 Report
the authors updated their manuscript accordingly.
Reviewer 3 Report
Authors clarified major point and introduced necessary amendments. The revised version can be accepted for publication.